# Reservoir Computing for Edge-based Automatic Speech Recognition

Nicolo Micheletti - 2024280039
Diego Cerretti - 2024280040
Thomas Adler  - 2024389002
*29 October 2024*

## 1   Introduction

Automatic Speech Recognition (ASR) ensures seamless interaction between humans and LLM-powered AI. Current state-of-the-art ASR models are transformer-based neural networks [1] that have a very high level of accuracy but come with the cost of high complexity, partly due to the attention mechanisms present in the model [1]. The latest ASR model by Open AI, Whisper Large, has 1.55bn parameters (2̃.9 GB) [11] and is reported by users [1] to require around 12 GB of VRAM to run. A model of this size has to be deployed on the cloud, which introduces network latency, slowing response times and degrading user experience. Indeed, edge devices cannot host a model of this size: the latest iPhone 15 Pro Max is estimated to have around 8 GB of RAM. Due to limited resources, current edge-based ASR models also struggle with accuracy [6]. Reservoir Computing offers the potential for a new generation of edge-based ASR models with low latency and high accuracy.

## 2   Background

Reservoir Computing (RC) [15, 9] is an ideal candidate for an edge-based ASR model because of its low training requirement and strong predictive capabilities in complex time series. The key property of an RC is that its internal layer is fixed, requiring no training. As a result, RCs require very little memory, computational power, fine-tuning or retraining. They can potentially be hosted on memory-constrained edge devices and avoid the network latency issues introduced by cloud-based models. RCs can achieve high-performance levels on specific tasks because of the rich non-linear and recurrent dynamics within the reservoirs. This enables them to extract complex spatio-temporal features when transforming input data into a higher-dimensional space. We further describe the RC architecture in Appendix A.

The following system of equations describes an RC.

$$\begin{cases} x(t+1) = (1-\gamma)x(t) + \gamma f(Wx(t) + W^{(in)}u(t) + b), \\ y(t) = W^{(out)}x(t), \end{cases} \quad (1)$$

It operates with discrete time steps, denoted by $t$. The non-linear activation function is represented by $f$, and the reservoir is the internal weight matrix $W$. $W^{in}$ is the input weight matrix, defining the nodes in the network that receive inputs (input-to-reservoir mapping), while $W^{out}$ represents the output weight matrix (reservoir-to-output mapping). The leakage rate $\gamma$ controls the amount of past information passed to the next time step. The input at time $t$ is denoted by $u(t)$, and $b$ is a bias term [15, 8].

---

[1] https://github.com/openai/whisper

## 3  Related Work

**Optimization strategies**  State-of-the-art ASR models achieve high accuracy but require complex architectures. Several techniques can be used to alleviate this issue. **Pruning** and **quantization** techniques can be combined to compress a model. Pruning cuts links in a dense network to reduce complexity, whereas quantization lowers the precision of parameters to save memory [7, 4]. Other methods have been implemented to improve convergence rates and training efficiency of ASR models, like **Sortagrad** and **automatic segmentation**. SortaGrad consists of initially training neural networks on shorter audio clips, and gradually increasing their length [2]. Automatic segmentation of input audio into meaningful units enhances the training of automatic speech recognition systems by improving how input features align with phonetic labels and simplifies the data the model needs to handle [5].

**ASR for Edge Devices**  Recent works have aimed to reduce the latency and memory footprint of ASR models to run on resource-constrained devices. Specifically, Gondi et al. [6] achieved this by building transformer-based models with quantization and other optimization techniques. Xu et al. [14] ran ASR on low-memory devices using Conformer CTC (Connectionist Temporal Classification Automatic Speech Recognition) models. Conformer CTC ASR is a speech recognition model that combines Conformer neural architecture, known for its efficiency in capturing both local and global speech patterns, with CTC loss, which aligns predictions with input sequences without requiring pre-labeled data alignment. Compared to traditional models, these systems exhibit an increase in Word Error Rate (WER), a common accuracy metric in speech recognition tasks.

**Reservoir Computing for ASR**  Recent works have attempted to reduce complexity while maintaining high accuracy in ASR using RC models. Picco et al. [10] implemented a photonic-based system that makes RC architectures suitable for high-dimensional audio processing tasks. Ansari et al. [3] used heterogeneous single and multi-layer RC models to create non-linear transformations of the inputs, capturing temporal context at different scales.

## 4  Proposal

Lighter models operating at the edge face challenges in balancing accuracy, latency, and efficiency. RC provides a promising alternative due to its lower inference latency and reduced memory consumption. In this study, we aim to harness RC to improve the performance of edge-based ASR.

For this project, we plan to evaluate our model on the English portion of the following two datasets:

- The *LibriSpeech*[2] corpus is a dataset comprising about 1,000 hours of audiobooks from the LibriVox project.
- The *Common Voice*[3] corpus is a multilingual collection of transcribed speech aimed at advancing research in ASR. The dataset contains 9,283 recorded hours.

We will compare our implementation against current state-of-the-art approaches in edge-based ASR: Whisper Small from OpenAI [11] and Wav2Vec 2.0 from Meta [6]. Both models performed well on ASR benchmarks but still struggled with accuracy compared to cloud-models, hampering their effectiveness.

To evaluate the performance of the models, we will use Word Error Rate (WER), Character Error Rate (CER), and frame-wise accuracy, representing the correct prediction of a sound present in a short audio segment. In addition, the models will be compared based on their memory usage and inference time.

To improve the accuracy of our RC model, we will explore various approaches, including those mentioned in Section 3. In addition, we will fine-tune RC-specific parameters. These include reservoir size and connectivity, the number of reservoir layers, recurrent weights, spectral radius, and the degree of chaotic dynamics within the reservoir [12, 16].

---

[2]https://paperswithcode.com/dataset/librispeech
[3]https://paperswithcode.com/dataset/common-voice

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

# A Reservoir Computing Architecture

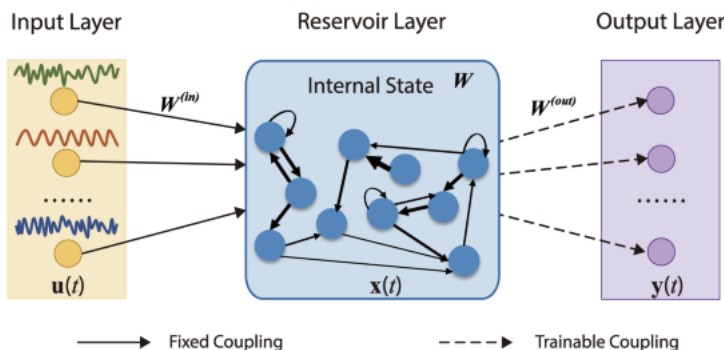

Figure 1: RC architecture [15] (Authorized under (CC BY 4.0)

As Figure 1 shows, an RC consists of three layers: an input (sensing) layer, a reservoir (processing) layer, and an output (control) layer. The input layer sends data to the reservoir, a fixed-weighted network that projects input data into a higher-dimensional feature space. The output layer, the only trainable component, typically uses linear regression to map these signals to the final output. The reservoir is initialized once, with its size, connectivity, and chaotic dynamics fixed. Then, the RC processes sequential data at every time step, and its recurrent connections ensure past data is carried over to future time steps [15, 13, 9, 8].

