# OpenReview forum: "Reservoir Computing for Edge-based Automatic Speech Recognition"
_tsinghua.edu.cn/THU/2024/Fall/AML — THU 2024 Fall AML Submission_

### Official Review · ~Aleksandr_Algazinov1 · 2024-11-06
**Well-explained, convincing, and relevant**

**Rating:** 10
**Confidence:** 4

**Review:**

The proposal is well-written and easy to follow. The authors do research on a rather new ASR problem. Based on various references, the authors proposed to experiment with a modern architecture. The project presents possible steps to improve the model, as well as a convincing motivation for the study.

---

### Official Review · ~Bowen_Su1 · 2024-11-08
**Comprehensive Proposal and Clear Methods**

**Rating:** 9
**Confidence:** 4

**Review:**

This proposal comprehensively reflects the author's thoughts and ideas. In response to the current problem of balancing efficiency and effectiveness in ASR, the author proposes using RC to improve the performance of edge ASR models.
The author provides a detailed introduction to the proposed method and further proposes evaluation criteria and detailed steps. Covered the requirements of the proposal.

---

### Official Review · ~Joydeep_Chandra2 · 2024-11-08
**Comprehensive new approach using RC in IoT and Edge domain but requires performance clarity on real life implementation**

**Rating:** 10
**Confidence:** 5

**Review:**

The focus on overcoming latency issues by using Reservoir Computing (RC) is timely and aligns well with the industry shift towards edge-based AI. The proposal provides a good overview of current state-of-the-art ASR models (e.g., Whisper and Wav2Vec 2.0) and optimization strategies. It shows a comprehensive understanding of the existing landscape and highlights the limitations of transformer models in resource-constrained scenarios. Although the proposal lists evaluation metrics like Word Error Rate (WER), Character Error Rate (CER), and inference time, it does not provide a clear plan on baseline comparisons or target benchmarks. More specific performance goals would help set expectations for the next phase.

---

### Official Review · ~Ziyad_Fawzy1 · 2024-11-11
**Reservoir Computing for Edge-based Automatic Speech Recognition**

**Rating:** 10
**Confidence:** 5

**Review:**

This paper explores Reservoir Computing (RC) as a solution to the limitations of current Automatic Speech Recognition (ASR) models on edge devices, which often struggle with high memory demands and latency when using transformer-based models. By leveraging RC’s low memory footprint and ability to capture complex temporal features, the authors propose an RC-based ASR model for edge applications, aiming to achieve high accuracy and low latency.

The authors clearly and rigorously explained the problem and their proposed method. Their work is practical and has potential in industrial applications.

---

### Official Review · ~Anqi_LI5 · 2024-11-11

**Rating:** 9
**Confidence:** 3

**Review:**

this proposal presents a promising and well-motivated research direction. Addressing the weaknesses mentioned above, particularly by providing more details on the approach and expanding the evaluation scope, would further strengthen the proposal and increase its potential for success.
Pros:
Addresses a critical gap in the field of ASR.
Clearly articulates the problem statement and research objectives.
Acknowledges related work.
Potential for high impact on the field of ASR.
Cons:
Lack of specificity in the approach.
Limited scope of evaluation.
Potential for overfitting.

---

### Official Review · ~Shuangyue_Geng1 · 2024-11-11
**Promising approach and well-designed structure**

**Rating:** 10
**Confidence:** 3

**Review:**

The proposal is novel, well-structured, and easy to understand. It presents a promising approach to improving edge-based Automatic Speech Recognition (ASR) by leveraging Reservoir Computing (RC) to reduce latency and memory usage compared to transformer-based models. It effectively highlights the need for low-resource ASR solutions and provides a solid overview of the current state-of-the-art. Moreover, it provides a comprehensive review of related work. Overall, this well-motivated research direction has the potential for significant impact.

---

### Official Review · ~Hector_Rodriguez_Rodriguez1 · 2024-11-11
**Review "Reservoir Computing for Edge-based Automatic Speech Recognition"**

**Rating:** 10
**Confidence:** 4

**Review:**

The authors make an excellent case for the use of Reservoir Computing for Automatic Speech Recognition (ASR) in memory-constrained devices.

- The introduction and background sections justify the need for lightweight ASR systems on edge devices.

- The related work highlights some approaches to optimize the resource utilization on ASR systems.

- The proposal section includes the datasets and other implementations that could be compared with the current RC-based approach. This section could benefit from a more detailed explanation on the RC architecture that would be implemented, as well as the target resource utilization and the desired inference time.

Overall, the proosal is well written and complies with all the requirements.

---

### Official Review · ~Michael_Hua_Wang1 · 2024-11-11

**Rating:** 9
**Confidence:** 4

**Review:**

Model size and computational requirements remain a perennial pain point and pose a major obstacle to broader adoption. The authors propose to apply reservoir computing (RC) to automatic speech recognition systems, seeking to reduce the resource requirements for a typical system while sidestepping the network-related latency induced by cloud computing-based approaches.

The proposal describes a plausible and novel application of this approach, and if successful, it can prove the viability of an approach to making models more accessible to consumers.

---

### Official Review · ~Zhuofan_Sun1 · 2024-11-12

**Rating:** 10
**Confidence:** 5

**Review:**

Strengths:
Relevance: The proposal addresses a crucial challenge in ASR, i.e., achieving high accuracy with low latency and resource consumption on edge devices. Reservoir Computing (RC) offers a promising solution due to its inherent properties.
Clarity: The proposal is well-structured and clearly explains the motivation, background, related work, and the proposed approach. The problem statement is clear, and the objectives are well-defined.
Feasibility: The proposal outlines a feasible plan with specific datasets, evaluation metrics, and comparison methods. The team also demonstrates an understanding of potential challenges and suggests approaches to address them.
Technical Soundness: The proposal demonstrates a solid understanding of RC and its applicability to ASR. The team acknowledges the importance of optimizing RC parameters for improved performance.
Areas for Improvement:
Specificity: The proposal could benefit from more specific details on the proposed RC architecture and optimization strategies. For example, what type of non-linear activation function will be used? How will the reservoir size and connectivity be determined?
Evaluation Plan: While the proposal mentions using WER, CER, and frame-wise accuracy, it would be beneficial to elaborate on the specific evaluation protocols, such as the training and testing splits, and the metrics for comparing memory usage and inference time.
Comparison Baselines: The proposal mentions comparing the RC model with Whisper Small and Wav2Vec 2.0. It would be helpful to justify the selection of these baselines and discuss any potential limitations.
Potential Challenges: The proposal acknowledges potential challenges but could provide more detailed discussion on how these challenges will be addressed. For example, how will the team ensure the robustness of the RC model across different accents and languages?
Overall, the proposal presents a compelling case for exploring RC for edge-based ASR. Addressing the areas for improvement will further strengthen the proposal and increase its chances of success.
Additional Suggestions:
Explore Different RC Variants: The proposal could consider exploring different types of RC, such as Echo State Networks (ESNs) or Liquid State Machines (LSMs), and compare their performance.
Investigate Transfer Learning: The proposal could investigate using transfer learning techniques to fine-tune pre-trained RC models on the target dataset, potentially improving accuracy and reducing training time.
Consider Edge-specific Hardware Acceleration: The proposal could explore utilizing specialized hardware accelerators for RC, such as FPGAs or neuromorphic chips, to further reduce latency and power consumption.
I believe this proposal has the potential to make significant contributions to the field of ASR and edge computing. I recommend further refining the details and addressing the areas for improvement to ensure a successful project.

---

### Official Review · ~Eddy_Yue1 · 2024-11-12
**Strong Proposal on Edge-based ASR Using Reservoir Computing**

**Rating:** 10
**Confidence:** 4

**Review:**

This project offers a compelling alternative to traditional ASR models for edge deployment. Addressing accuracy constraints may help ensure practical applications across various edge devices.

Related work section is thoroughly researched, overall detail in proposal is convincing.

---

### Official Review · ~Justinas_Jučas3 · 2024-11-12
**Extremely Clear and Well-Structured proposal**

**Rating:** 10
**Confidence:** 4

**Review:**

In general, the proposal is extremely concrete and clear to understand. The significance of the work is also important and described in the proposition. All of the basic requirements are fulfilled. In addition, I would say that the project also seems feasible to be implemented within limited time constraints, which makes it realistic.
## Advantages
1. Very clear and well-structured report
2. Satisfies all of the requirements
3. The proposal is ambitious, yet seems feasible to be implemented

## Disadvantages
1. Perhaps the Related Work/Proposal sections lack some discussion on the potential limits of the RC, as these obviously exist. However, this is the only one I could think ok.

---

### Official Review · ~Chendong_Xiang1 · 2024-11-12

**Rating:** 8
**Confidence:** 2

**Review:**

This paper explores using Reservoir Computing (RC) for Automatic Speech Recognition (ASR) on edge devices, addressing the challenges posed by traditional ASR models that require high memory and computational power, such as those based on transformers. Unlike these models, RC has a fixed, untrained internal reservoir layer, allowing efficient processing of time-series data with minimal resource requirements. This makes it well-suited for memory-limited edge devices.

The study tests an RC-based ASR model on datasets like LibriSpeech and Common Voice, comparing it to smaller transformer models (e.g., Whisper Small) in terms of Word Error Rate (WER), Character Error Rate (CER), and inference time. By fine-tuning RC parameters, such as reservoir size and connectivity, the authors aim to enhance accuracy without compromising efficiency.